# Holistic Security and Safety for Factories of the Future

**DOI:** 10.3390/s22249915

**Published:** 2022-12-16

**Authors:** Eva Maia, Sinan Wannous, Tiago Dias, Isabel Praça, Ana Faria

**Affiliations:** GECAD—Research Group on Intelligent Engineering and Computing for Advanced Innovation and Development, School of Engineering of the Polytechnic of Porto (ISEP), 4249-015 Porto, Portugal

**Keywords:** smart factories, cybersecurity, Industry 4.0, multi-domain security and safety, digital twins, security of Factories of the Future

## Abstract

The accelerating transition of traditional industrial processes towards fully automated and intelligent manufacturing is being witnessed in almost all segments. This major adoption of enhanced technology and digitization processes has been originally embraced by the Factories of the Future and Industry 4.0 initiatives. The overall aim is to create smarter, more sustainable, and more resilient future-oriented factories. Unsurprisingly, introducing new production paradigms based on technologies such as machine learning (ML), the Internet of Things (IoT), and robotics does not come at no cost as each newly incorporated technique poses various safety and security challenges. Similarly, the integration required between these techniques to establish a unified and fully interconnected environment contributes to additional threats and risks in the Factories of the Future. Accumulating and analyzing seemingly unrelated activities, occurring simultaneously in different parts of the factory, is essential to establish cyber situational awareness of the investigated environment. Our work contributes to these efforts, in essence by envisioning and implementing the SMS-DT, an integrated platform to simulate and monitor industrial conditions in a digital twin-based architecture. SMS-DT is represented in a three-tier architecture comprising the involved data and control flows: edge, platform, and enterprise tiers. The goal of our platform is to capture, analyze, and correlate a wide range of events being tracked by sensors and systems in various domains of the factory. For this aim, multiple components have been developed on the basis of artificial intelligence to simulate dominant aspects in industries, including network analysis, energy optimization, and worker behavior. A data lake was also used to store collected information, and a set of intelligent services was delivered on the basis of innovative analysis and learning approaches. Finally, the platform was tested in a textile industry environment and integrated with its ERP system. Two misuse cases were simulated to track the factory machines, systems, and people and to assess the role of SMS-DT correlation mechanisms in preventing intentional and unintentional actions. The results of these misuse case simulations showed how the SMS-DT platform can intervene in two domains in the first scenario and three in the second one, resulting in correlating the alerts and reporting them to security operators in the multi-domain intelligent correlation dashboard.

## 1. Introduction

The introduction of the Industry 4.0 paradigm has introduced a digital revolution of IT-driven aspects into industrial processes. Concepts such as the Internet of Things, smart factories, and decentralized self-organization have been at the heart of this technological revolution [1]. In this sense, the aim of the Industry 4.0 shift was to level up the automation of factories by leveraging the capabilities of advanced sensors, embedded software, and robotics. As a complementary paradigm, the Fifth Industrial Revolution (a.k.a. Industry 5.0) [2] has been initialized to make efforts toward a sustainable, human-centric, and resilient European industry. This new industrial movement aims at achieving societal goals beyond jobs and growth. That being said, Industry 4.0 and 5.0 are attempts to capture the maximum value of new technologies while placing the well-being of the industry workers at the center of the production process [3,4].

Additionally, the introduction of modern technologies to realize smart factories has led to an increased intersection between cyber and physical components. The goal of Industrial Cyber-Physical Systems (ICPS) is to combine engineering models and methods to manage critical infrastructures on the basis of data captured from edge sensor networks. From a security standpoint, the coexistence of cyber and physical counterparts in a networked structure opens the doors for additional attacks and vulnerabilities to endanger future industrial systems. This may lead to losses of economic benefits or disorder of social life [5]. In this regard, Intrusion Detection Systems (IDSs), Network Traffic Analysis (NTA), and anomaly detection techniques are being used to mitigate security concerns [6].

In a related subject, the increasing connectivity in the Factory of the Future (FoF) environments is usually accompanied by greater amounts of related security risks [7]. This impact is enlarged when considering industrial systems that have never been technically designed for a networked environment. Nevertheless, besides the security challenges this interconnectivity poses to smart factories, it brings opportunities to address a variety of threats and safety issues. The research problem our work aims to cover is to investigate the effectiveness of correlating existing alerts to detect new ones. Such individual alerts being generated separately might not indicate an abnormal situation, while if seen together would help security operators to gain insights about dangerous situations in the industry. In this regard, the possibility to accumulate and analyze activities taking place simultaneously in different parts of the factory is a key enabler to such opportunities, primarily by correlating large amounts of data, gathered from various sources, in order to establish cyber situational awareness of the investigated environment. Such a kind of integrated security overview would potentially allow for the adoption of suitable countermeasures in case of attacks [8]. Multiple approaches have been discussed in the literature to promote security and safety in smart environments. Some architectures adopt multi-agent and multi-sensors systems to detect emergency situations in intelligent buildings [9], industrial plants [10], and smart factories [11,12]. Other proposed approaches extend the potentials of data correlation to support security operators in collecting relevant data from various sources [13,14,15]. To this end, inspecting seemingly unrelated events and correlating them in a smart industrial context is crucial to obtain a holistic security and safety overview of the factory of the future.

Additionally, as a promising enabling technology to realize smart manufacturing, Digital Twin (DT) is characterized by the cyber–physical integration that yields making accurate predictions, rational decisions, and informed plans [16]. In manufacturing, digital twins have attracted interests from big industrial players and are usually used to predict and inform about equipment failures and non-optimal performance, as well as to improve customer experience [17].

In the context of industry digitalization, factories invest in enterprise resource planning systems (ERP) to manage their whole activities. These integrated tracking systems possess great potential to increase efficiency, security, safety, and resilience of the targeted industrial environments. Moreover, the ability to provide data and knowledge as a service is a critical asset to answer the market needs and achieve business transformation. The main contribution of our work is to provide knowledge as a service and introduce cyber-security awareness into industrial ERP systems. This takes place by utilizing an increased set of services, along with environmental and network sensors, in order to assure the safety, security, and resilience of manufacturing systems, thus achieving a holistic optimization and intelligent decision support of factories.

The application field of our work is a textile manufacturer, wherein the main assets are specialized human resources, looms, and cutting machines. In this context, multiple sensors have been installed and utilized to measure environmental conditions and monitor machines parameters. A digital twin framework was employed to mirror the cognitive status of various entities including network, employees, and equipment. A data lake has also been used to store collected information, to easily access the data, and to analyze it to uncover abnormal patterns. Furthermore, we developed a set of intelligent services on the basis of innovative data lake analysis and learning approaches, including (a) services for aggregation of information from the real-time monitoring; (b) analytic services for the efficiency and efficacy of data usage based on hybrid methodologies; (c) intelligent decision support to transform the available data into knowledge that support decision making; and (d) prediction applications and mechanisms for stream and complex event processing. Additionally, human behavior was also modelled and simulated on the basis of affective computation to investigate stress, fatigue, and lack of attention in the interactions between humans and machines.

All the mentioned systems are combined in the integrated SMS-DT platform that we developed to simulate and monitor realistic industrial installations in a DT-based architecture. Figure 1 depicts the key capabilities and envisioned schema of our systems. With this novel knowledge module and integration, ERP systems will be able to derive the best operational conditions, as well as to optimize and automatically control the resources, taking into account the real-time, or near real-time, situational awareness of the complete factory environment (i.e., cyber-physical systems, materials, social and economic variables).

The rest of this document is organized as follows: In Section 2, we list similar previous and ongoing activities. Section 3 presents a brief description of digital twins and how they can be implemented in industrial environments. Section 4 and its sub-sections illustrate the different dimensions of our SMS-DT platform, including developed systems and intelligent services. Section 5 describes the textile industry use-case as well as two misuse-cases that can be discovered and prevented using the SMS-DT platform. Finally, we conclude our work in Section 6.

## 2. State of the Art

Cyber security threats are major issues that affect the sustainability of manufacturing in Industry 4.0 [18]. This is primarily because malicious software or virus attacks can access different machines and affect the manufacturing activities running on the same network infrastructure. Similarly, the transition to more open network architectures brings new security challenges to smart factory systems, primarily due to the huge amounts of connecting IoT devices such as sensors, actuators, and edge-computing units, along with other end devices in automated information networks [19]. In this regard, most previous and ongoing activities try to envision general architectures to simulate and optimize processes in smart factories. Others provide surveys to investigate security issues in modern industrial systems. However, we identified a relatively limited number of studies that focus on the integrated security of smart factories as a critical factor to secure persistent and risk-free production conditions. We highlight the most relevant studies to our work in the following paragraphs.

Studies have been conducted to leverage the potential of new technologies to generate useful insights and achieve safer and more secure industrial environments. The authors of [11] describe a machine-learning-based and context-aware intrusion detection system in a smart factory environment. This system architecture has three main phases: (i) data capturing from sensors and resources, (ii) model building and inferring using unsupervised/supervised models, and (iii) threat visualizing in a real-time representation. The results of this study showed better detection rates for anomaly signs. Additionally, the authors of [12] worked on maintaining security over large data-driven systems in Industry 4.0. Thus, they proposed a new threat intelligence scheme that models the dynamic interactions of physical and network components. It consists of two modules: smart management and threat intelligence. The first module handles heterogeneous data captured from sensors, actuators, and network traffic, while the second one discovers anomalous activities in physical and network systems. The scheme was evaluated, and it outperformed other peer mechanisms. In another context, the authors of [10] state that due to the huge amount of involved variables, the selection of alerts in industrial settings tends to be difficult. Thus, the authors resorted to semantic web and machine learning techniques to propose an industrial-context-aware recommendation system. The solution has an adaptive interface to make non-intrusive recommendations to facilitate the decision-making process. The system was evaluated using a real database and showed promising results.

On the other hand, the rapid technological development and emergence of new digitization capabilities have promoted the realization of smart factories. This requires leveraging new design methodologies that rely in the first place on data-driven and AI-assisted approaches. Digital/virtual factory and smart/cloud manufacturing are some of major trends in the digitalization of a Factory of the Future that were compared and analyzed by [20]. On the basis of this analysis, the building blocks of the Factory of the Future were proposed in the form of a reference framework for the realization of a digital twin model. The framework consists of various interconnected components such as IoT technologies, cloud manufacturing (private, public, hybrid), factory models (smart, digital, virtual), and multiple forms of infrastructure as service modules. According to the authors, there are multiples challenges in this context, primarily in terms of model accuracy and fidelity that requires higher degrees of synchronization between virtual and real shop floor entities.

IoTwins is a European project that works on facilitating the implementation of Industry 4.0 technologies to increase productivity, safety, resiliency, and environmental impact [21]. The project leverages the concept of hybrid DTs to create reference architectures for production plants and processes. Hybrid DTs mix both data-driven and model-driven approaches. The data-driven aspect refers to cloud-based data analysis and machine learning models, while the model-driven aspect represents software simulators that mimic the behavior of physical assets in connection with the data-driven models. The proposed architecture of the IoTwins project is described in [22] with multiple layers: IoT, edge, cloud, and application layers, along with authentication and ancillary services. In this research, an industry-related use case was presented to show how to leverage this platform to execute distributed DTs for predictive–maintenance purposes. The use case collects different application requirements and, as a result, proposes a solution that includes front-end, back-end, and production monitoring tools and components. COGNITWIN is another European project that sets standards for the design, development, and operation of the European process industry [23]. This project introduces hybrid and cognitive digital twins through four big data and AI pipeline steps adapted for digital twins. In relation to this project, the work by [24] presents the DT pipeline framework of the COGNITWIN project by illustrating the use of a hybrid DT approach with a focus on the DT support for predictive maintenance. The presented approach aims to reduce energy consumption and average duration of machine downtimes. The authors concluded that the pipeline can be used for similar cases in the process industry. Furthermore, hybrid DTs have been also discussed as an enabling technology for the fourth industrial revolution. The study by [25] discovers the combination of virtual reality, 3D imaging, and hybrid DTs for factory layout planning. In this work, three industrial case studies were presented by utilizing the hybrid digital twin models. The results showed that using such models, discovering implicit knowledge can improve solution fidelity, and multiple issues during planning and installation phases can be avoided. In a later publication, the authors presented the combination of 3D laser scanning, virtual reality, CAD models, and simulation modelling in a hybrid digital twin [26], wherein the results also showed that the planning process can be noticeably improved, yielding benefits in all phases.

In a related subject, the platform of smart factories requires having both workers and robots working simultaneously in one environment. According to [27], the collaboration between humans and robots in industrial settings can be hazardous. In this paper, the authors present a systematic review of the safety concerns of human–robot collaboration (HRC). According to this study, HRC systems can be tremendously complex, and the real-world AI systems including HRC applications should be designed to able to prevent unintended or harmful behavior. Moreover, AI-based (smart) robotic systems are strongly becoming one of the main areas of focus in manufacturing towards future factories. In a similar work, the authors of [28] investigated the safety of workers in a factory shop floor. The authors used sensors to gather normal and abnormal data of human activities at the factory, in which a real-life situations dataset was also developed.

To this end, and with respect to the scanned literature, the majority of conducted studies contributed to the security of smart factories from a partial or domain-specific angle. However, we were unable to find relevant efforts that try to approach the security issues from a holistic and inter-domain perspective. The work presented in this paper is an extension to our previous multi-domain security awareness for Factories of the Future [29]. Primarily, it brings enhancements in terms of leveraging the capabilities of factory simulations and digital twins to create a digital representation of the factory components. The approach we are describing here is totally new in terms of promoting security monitoring in smart industrial settings by capturing and correlating seemingly unrelated alerts from multiple domains (network, human behavior, energy consumption, etc.). Such alerts may seem to be quite normal or not critical if seen individually. However, they might be able to detect dangerous scenarios if correlated with other events occurring simultaneously in other domains. In this paper, we describe our envisioned architecture, its integrated systems, and the intelligent correlation components.

## 3. Digital Twins in the Factories of the Future

Several definitions of digital twins exist in the literature, and each author adapts the definition according to their purpose. However, all the definitions have something in common. Generally, a DT can be defined as a simulation of something in a specific environment [30]. Digital twins in manufacturing have been widely discussed in the literature. Several works were conducted to survey, review, and classify digital twins concepts, applications, and technologies from different perspectives [31,32,33,34,35]. Conversely, others proposed DT-based knowledge-driven framework and systems architectures to simulate data exchange in physical and digital layers in smart manufacturing [36,37,38,39,40,41].

In the context of the manufacturing sector, the ability to simulate all industry processes, along with an increasing level of digitalization and complexity of machines and products, will offer faster, safer, and less expensive modifications to existing and future asset/system implementation. Moreover, since DTs can run in parallel to their physical counterparts, changing specific parameters in the production process, adapting capacities, or using different raw materials can all be simulated using a DT. Therefore, for the factories of the future, digital twins allow them to increase efficiency and productivity, supporting companies in monitoring plant construction, managing assets, and testing their end products [42].

The ability of DTs to run synchronously with their physical parts is also crucial to ensure the safety and security of the FoF systems [43]. Using DT simulation, it is possible to perform deep inspections of their behavior without the risk of disrupting the operational technology services. Using intelligent services, it is possible to track data inconsistencies between the physical and the virtual parts of the systems in order to reduce unpredicted and undesirable issues caused by several factors, such as network failures or erroneous human interactions [44].

The human dimension in FoF is also of crucial importance, not only from the manufacturing optimization point of view but also in terms of fostering security in FoF. However, this area is in a very early stage of development. Moreover, most of the efforts being made in this area are only interested in modeling human behavior for an ergonomic evaluation or even a human simulation. However, validation of the models, collection of real worker data to provide model inputs, comparison of the simulation results with the manufacturing systems, and sensitivity analysis are significant innovation fields where further research is needed [45].

In the context of our work, we aimed to use real worker data to develop models that allow its inclusion in the FoF modeling activities through DT development. Thus, our system interprets and mirrors human emotions/states, such as fatigue, lack of attention, or stress. For that, not only operator facial expressions are analyzed but also, using affective computing, the events generated from the user interaction with terminals are considered. Additionally, our model also simulates the complete cyber-physical manufacturing systems. This way, we aim to overcome the usual inability of DTs to comprehensively inform decisions, not only at the individual machine level but also in the entire production ecosystem. Sensor data collected from machines and the shop floor environment will be analyzed in our SMS-DT, thus allowing a quick reaction to anomalies and fast recovery processes. Our system aims to analyze a large set of heterogeneous data and to explore data dependencies, thus providing unexplored perspectives of the shop floor. The SMS-DT system learns from data collected on the shop floor with several models implemented, for example, to forecast the energy consumption of equipment and the consumption from the different sections of the factory.

The modeling of human behavior, detection of abnormal network traffic or behaviors, and energy modeling and consumption forecasting are key features of our SMS-DT system. It not only detects abnormal behaviors from the technical, environmental, and human dimensions, but it also raises alarms and alerts, expected to reach the right person at the right time. The SMS-DT system avoids intrusive mechanisms in the operator’s routine so that our assessment does not become an influencing factor in the system, thus avoiding any impact on production.

Technically speaking, many companies do not have the expertise or resources required to effectively implement digital twins. This becomes even further complicated considering the lack of consensus on unified concepts, standards, technologies, and procedures [41]. Several software solutions exist to implement digital twins; however, most of them are focused on only one specific digital twin specification. Moreover, the main solutions are proprietary software designed to fulfill the company’s product aim. From the few open-source solutions available, we have identified Eclipse Ditto and ThingsBoard as the most interesting ones. ThingsBoard [46] allows for the collection, processing, and data visualization of many devices. It generates alarms through a rule chain engine. On the other hand, Eclipse Ditto [47] is designed for the Internet of Things. It holds models of “things” as JSON objects, and it has an API that supports HTTP, WebSocket, and other industrial protocols such as MQTT.

## 4. SMS-DT Platform

Several reference architecture models have been developed in recent years to address the issues of integration and interoperability in the context of smart manufacturing. The initial architecture modeling focused strongly on the business and operational levels, while newer approaches tried to also include architectural aspects of the involved systems. 1990’s PERA (Purdue Enterprise Reference Architecture) [48] is an example of the first category, which aims at computer-integrated manufacturing. More recent architectures have been created within Industry 4.0 initiative, with the models proposed by Platform Industrie 4.0 [49] and Industrial Internet Consortium (IIC) [50], two of the largest organizations that research topics related to industry and industrial internet, respectively. Other prominent architectures include the ENISA Purdue Model [51], IBM Industry 4.0 [52], and NIST Service-Oriented Architecture [53].

To describe the SMS-DT platform, we decided to follow the IIRA architecture model based on ISO/IEC/IEEE 42010:2011 and presented by the Industrial Internet Consortium (2017) [54]. It was built as an open architecture with a wide array of applications across industries. The term industrial internet is used to represent the Internet of Things, machines, computers, and people, enabling intelligent industrial operations using advanced data analytics for transformational business outcomes [55]. Figure 2 illustrates, using the three-tier architecture, the different components of SMS-DT. The three-tier architecture comprises three different tiers that play specific roles in processing the data flows and control flows involved in usage activities: edge, platform, and enterprise tiers. The edge tier collects data from the edge nodes using the proximity network. The architectural characteristics of this tier are dependent on the specific use cases. The platform tier is used to send the control commands from the enterprise to the edge tier. It also receives data that are ingested, organized, and then stored in the appropriate place. Non-domain-specific services such as data query and analytics are also possible in this tier. Optionally, it may also provide management functions for devices and assets. The enterprise tier carries out industry domain-specific business applications and related decision support systems. It also provides interfaces to businesses end-users and operation specialists. This tier often receives data flows from the edge and platform tier. It also issues control commands to the platform and edge tier.

Considering the SMS-DT platform, on the shop floor or edge tier, several events are tracked: equipment, sensors, humans interacting with each other, humans interacting with robots and terminals, robots that communicate with each other, etc. Information such as equipment state, energy sensor measurements, human facial images, human interaction with mouse and keyboard, and network traffic can be captured and sent to a data lake in the platform tier, where all information is stored. A data processing module acts as a data broker, feeding and receiving data from the other modules. The data are pre-processed according to the data modality, e.g., filtering noisy data on an image or audio record, or missing records from a sensor are removed or replaced by default values. Furthermore, in multi-modality approaches, the data are aggregated from different data sources and stored in a data lake. The digital twin module mirrors the cognitive status of the employees, energy, and network state. Using intelligent techniques, these different modules monitor, detect, and predict issues in the system. As a result, they send alerts to a multi-domain component that correlates all the alerts to obtain universal monitoring of the occurring events. The correlator also assists in establishing cyber situational awareness and adopting suitable countermeasures in case of attacks. All the information in the different modules can be visualized in different dashboards, aggregated in a common visual interface at the enterprise tier. Here, it is possible to display a summary of the different events and the alerts generated in real time.

### 4.1. Sensing and Monitoring the Shop Floor

SMS-DT comprises a digital twin that allows for the emulation of the equipment on the shop floor and its surrounding environment. A brief overview of the digital twin architecture can be seen in Figure 3. It is based on Eclipse Ditto, an open-source digital twin application that allows for the storage of information regarding a ”thing” or equipment. Ditto can receive information from sensors or equipment by HTTP or MQTT protocols and make it available through a REST API. It stores the thing’s information as JSON objects, allowing any change to be made to these objects, even partially, using versatile access URLs in the REST API. Ditto was chosen because of its open-source nature and flexibility. The only other comparable open-source software is ThingsBoard, which we found to be less flexible. Changes are quickly made on Ditto software, with a simple call to the REST API, while in ThingsBoard it is hard to deal with the changes. In our implementation, Eclipse Ditto ran as a multi-container docker through the tool Docker compose.

The sensor information is read from the data lake through the Node-RED application. Node-RED is a Node.js-based relay application for wiring together hardware devices through messages exchange, which we used as an information broker. It reads sensor data and inserts it in bulk in our database. The information gathered from the sensors is then stored in ElasticSearch, a highly scalable NoSQL database with a powerful search and analytics engine. ElasticSearch along with Kibana and Logstash composes the open-source ELK Stack [56]. We used ElasticSearch to store data, such as sensor readings and test results, in a way that can be scaled for an industrial environment. Kibana is a data exploration and visualization tool that runs on top of the ElasticSearch database. We used it to provide the equipment data visualization dashboards for the digital twin. A graphical user interface was developed, and Kibana dashboards were embedded in order to assure easy access to all this information.

As a prototype, we deployed a partial digital twin of three pieces of equipment that mirrors the equipment state information. The equipment DT visual component is composed of two pages, a map that allows for the selection of equipment and viewing basic information Figure 4, and the equipment panel (Figure 5).

The equipment panel displays not only the state information stored in the DT software (Ditto) but also two dashboards. One dashboard shows gauges with the current average sensor data from the last twenty seconds, while the other shows historical sensor data from the last hour in the form of graphical timelines and maximum and minimum values, as can be seen in Figure 6.

Since we were dealing with important shop-floor data, we also had an authentication page to ensure that only allowed persons to have access the application. Keycloak [57], an open-source authentication software that supports Open ID connect, was used for that aim since it allows user information management and provides authentication and authorization capabilities.

### 4.2. Human Behavior Dimension

Human resources are a crucial asset of any factory. The performance of each worker during the working day greatly affects the factory’s production. Moreover, a stressed or fatigued worker is more likely to mis-operate machines or make mistakes when dealing with them. A lack of attention during the job can affect the continuity of the factory’s processes or even cause injuries to the workers and/or co-workers. In this context, it is important to understand if these mistakes are accidental or malicious. To determine this, understanding human intention by comparison with normal behavior patterns is required. Therefore, it becomes important to monitor the operator’s behavior during work to avoid unexpected and unpleasant issues.

Fatigue is one of the most important operator states to be detected, since it greatly impairs the operator’s performance, especially in industry settings. There are several methods and/or technologies currently being used to detect fatigue and other associated states, namely, visual facial detection, keyboard and mouse interaction patterns analysis, and mental chronometry. In the context of textile factory circumstances, we believe that mental chronometry is one of the useful techniques to be implemented. Mental chronometry is the study of response time or reaction time (RT) in tasks [58]. It can be used to infer fatigue levels since it has been demonstrated that with the onset of fatigue, reaction time decreases, and the number of errors increases [59]. Visual tasks such as Go/NoGo testing can be used with success to measure response inhibition in these factors [60]. In a Go/NoGo task, a subject must perform repetitive motor responses to two stimuli, Go and No-Go. In the Go stimulus, the subject is normally presented with a visual stimulus that requires the worker to press a button. In the No-Go stimulus, the subject is required to inhibit the usual go response, abstaining from performing an action. Stroop interference is another task where one is required to suppress a dominant response [61].

Therefore, we developed a Go/NoGo test framework that can be accessed by the digital twin interface presented in the previous section (Figure 7). On a normal textile shop floor, it is usual that operators need to react to some machine alarms. Thus, our aim was to redirect the operators to this testing framework if an anomalous reaction time to this alarm was detected. The framework developed has two variations of the tests. First, a basic Go/NoGo test is triggered. Then, on the basis of the results, the framework can question the user about their fatigue level using a USAFSAM fatigue scale [62]. We think that providing workers with early warnings about their fatigue level can allow better fatigue management and thus improve the operator’s performance.

Additionally, the user interaction patterns with the ERP system as well as peripheral devices (keyboard/mouse/screen) are captured and analyzed as they assist in the detection of fatigue and other associated states in operators. For example, the analysis of micro-expressions in the operator’s face or sudden changes in the operator’s speech are good indicators of the operator’s performance. Not only stress and fatigue can be detected with this analysis, but also malicious actions of the operators can be recognized. Several technologies and solutions have been proposed with success in this field, such as the analyses of physiological signals, speech signals, and human facial images [63].

In this context, we decided to study how facial emotion recognition [64] and speech [65] can be used to detect and classify the subject’s emotions. From our study emerged the Human Behavior Analyzer, a tool that we developed to monitor the emotions of shop floor workers. This module captures and analyzes behavioral signs via two integrated components: the camera application (CA) and the emotion detector (ED). The former keeps track of the operator’s emotions and records real-time footage of the user via a webcam. The latter classifies the emotion expressed in the images captured by the webcam. Both components are implemented in Python using frameworks such as PyQT, FastAPI, and OpenCV. As described in Figure 8, the CA captures footage of the operator and sends it to the ED. In turn, this component starts the classification process, which consists of three processing and one classification phases: (i) image preprocessing; (ii) person identification; (iii) face identification; and (iv) emotion classification.

In the first phase, the received image is converted to RGB format and normalized by dividing the three colors channels by 255. The preprocessed image then enters the second phase, which consists of identifying the primary person in the frame. Resorting to object detection methods, we used the ssd_512_mobilenet1.0_coco pretrained algorithm made available by GluonCV [66] to perform the person detection. Only the person with the highest detection percentage is considered for the analysis. The third phase takes the pose outputted from the pretrained algorithm and uses it as input to simple_pose_resnet18_v1b pre-trained algorithm, made available by the same framework as an object detector, which provides facial coordinates of the identified person. This information is then used to crop the image in order to avoid misclassifications caused by the surrounding environment. Lastly, the facial image of the operator is then classified by an EfficientNet Convolutional Neural Network [67] trained on an FER2013 dataset [68]. This module considers seven emotions based on the Ekman Model of Basic Emotions [69]. These are angry, surprised, disgusted, happy, fearful, sad, and neutral. Upon emotion classification of the operator, ED correlates the image output with the IP address of the source PC where the image was taken from, thus preserving the operator’s public integrity and confidentiality. The results are then returned to the CA and are also published to the data lake to be consumed by other subscribed modules in the SMS-DT platform. On the basis of the assessment of streamed frames, the CA suggests taking fatigue tests in case the majority of obtained emotions are negative for a long period of time, as shown in Figure 8.

### 4.3. Network Dimension

Intrusion detection systems have been largely used to identify and deal with different threats, primarily by analyzing network traffic and reporting eventual unnatural occurrences. Suricata [70] is an example of an open-source IDS that supports several protocols and has several rulesets that can be easily extended. Even using several rules to filter the network traffic received, the IDSs deal with large amounts of diverse data. Machine learning is particularly useful at dealing with large and varied datasets, which are crucial to developing an accurate intrusion detection system. Thus, the huge challenge that intrusion detection represents can be supported by machine learning techniques.

Therefore, we built a machine learning engine that works together with Suricata IDS to identify different threats. The ruleset of the Suricata IDS instance used was extended with rules designed according to the investigated shop floor to match its specific threats and anomalies. This ruleset is completely flexible and allows the administrator to add new detection rules. Each rule consists of an action (what happens when the signature matches), the header (protocol, IP, ports, direction), and options. When a match with a rule occurs, an alert is raised (according to the action defined). The ML engine was developed to help in the detection of new attack patterns and new vulnerabilities. The structure of this engine is described in Figure 9. The ML engine receives the network traffic and uses anomaly detection (unsupervised learning) and misuse-based (supervised learning) models to detect attacks and anomalies. Then, this information is outputted to help in the detection of security incidents.

The ML creator is the core of the ML engine. It is responsible for continuously creating new models to improve the performance of the ML engine. This is very important since due to the fast emergence of new attacks, data need to be constantly updated. The creation process consists of three main stages: pre-processing, training, and evaluation, as can be seen in Figure 10.

Pre-processing: in terms of traffic monitoring, most raw traffic network information is not prepared to be fed into machine learning algorithms. As such, most algorithms require some processing of the data to function properly, e.g., removal of 0 variance features, or removal of invalid values. Moreover, some parsers also need to be built to ensure the match with machine learning algorithm input. Feature selection and feature engineering are techniques that aim to reduce the number of variables in the data to remove noise and accelerate the models. They are also used to create new features that have better relationships with the target variables.

Training and evaluation: the training process occurs after carefully splitting the available data into train and test sets and putting away the testing set for later use. If the data are enough, the train set is further split into a validation dataset that is used to evaluate every model trained in the training set until a performance goal is achieved. In the last instance, the model is tested on the test set, and if the results are satisfactory, the model is deployed. In the case of data shortage in the training set, the model can be trained using cross-validation, where the training set is split into k parts or folds, and then the model is tested independently on each of the parts while being trained on the aggregation of the other k-1 parts. The performance is then calculated as the average of the performance of all folds.

In the first stage, we performed several experiments using public datasets. Some unsupervised learning techniques, which may hold the key to the detection of zero-day attacks, were studied. Additionally, feature selection and ensemble methods were applied to recent datasets to develop valid models to detect intrusions as soon as they occur. A robust and simplified framework that allows for the optimization of any machine learning model was also built. All these works were developed with the aim of improving the intrusion detection capability. Note that these experiments corresponded to a first phase of the continuous improvement of the ML engine. In this phase, we used the publicly available datasets that are intended to resemble real data traffic to create the first models to be included in the ML engine. We used these datasets since they have several different attacks, typically used in real networks, that allow machine learning algorithms to learn how to detect them. Then, the ML engine was fine-tuned with real data from the actual network of the inspected shop floor.

### 4.4. Optimization Dimension

One of the main capabilities of digital twins in smart factories is to simulate the processes and situations of both physical and digital components. Simulation-based modules might help in monitoring, assessing, and analyzing the performance of multiple aspects in a manufacturing environment. Intelligent-based simulation tools can be developed not only to promote decision making in smart factories but also to optimize the ongoing processes and help in avoiding unseen and undesired behaviors.

Due to its vital role and contribution in almost all components of modern industrial systems, monitoring power consumption in FoF is of paramount importance. It is crucial to secure a solid overview of the different parts of the factory and their current and predicted behavior. In this context, we developed energy forecasting tools and analyzers to provide dynamic forecasting services for buildings and factories. The aim of our tools is to compare and build forecasting models using machine learning algorithms and historical data sets. They produce predictions for short- and medium-term horizons for the observed entities, in essence by utilizing historical data sets with the most common input features, such as previous consumption values, weather data, and time contextual fields (hour, day of the week, day of the month, month of the year, year, etc.). Technically speaking, the tools are standalone web applications that deliver their services via interactive graphical user interfaces. The energy forecasting tools are configurable and built in Python using Django web framework and Scikit-learn library [71]. Some prediction services are also provided remotely using Restful API services. The tools utilize multiple machine learning models including Adaboost.R2 [72], random forest [73], gradient boosting regressor [74], support vector regressor [75], and linear regressor [76]. They also provide a variety of dynamic energy forecasting services from model training, prediction, and tuning to other customized services.

Furthermore, one of the key features of the developed tools is to analyze the actual power consumption for a set of observers/sensors in real time. Using machine learning models, the energy analyzers predict future energy records for each observer, compare actual and predicted values, and generate forecasting-based alerts. To achieve this goal, the analyzer runs through two periodical phases. The training phase is when a training scheduler for each observer is set and triggered. In this phase, the latest historical energy consumption data for the observer are retrieved and pre-processed. Then, they are aggregated into a specific timestamp and used to train a supervised ML model, resulting in an up-to-date trained prediction model for each subscribed observer. We designed an input data adapter to extend the energy analyzer to retrieve power records from multiple channels, including APIs, databases, and static MS Excel files. On the other hand, the scanning phase is when a scanning scheduler for each observer is defined and triggered. During this process, the actual energy consumption for the targeted observer is retrieved. The previously trained model is used to predict the expected consumption. Then, the analyzer compares both predicted and actual consumptions and assesses the difference. If the assessment found that the actual consumption exceeds the normal predicted one, an alert is triggered and sent to a broadcasting channel. For this aim, we designed an alert triggering mechanism to allow broadcasting alerts to multiple channels such as APIs, databases, and Apache Kafka topics. Figure 11 shows the JSON structure of an alert generated and broadcasted by our energy forecasting analyzer from the textile shop floor.

In our context, all alerts being generated and triggered by the energy analyzers are sent to a Kafka topic so that they can be consumed by other modules in our architecture and correlated with events taking place in other parts of the investigated space.

### 4.5. Multi-Domain Dimension

In the FoF environment, it is essential to develop mechanisms that ensure the maximum availability of production assets in crisis situations. An important issue to be considered is the correlation of the information from different sources in the FoF environment, namely, information about shop floor monitoring (energy consumption analysis and human behavior monitoring) and information about intrusion detection (network behavior monitoring). Figure 12 shows an overview of the correlation of the investigated shop floor information. As can be observed, different cyber and physical events on the shop floor are collected. These events are sent through the Kafka engine to a multi-domain correlator that will work on the correlation. Apache Kafka is a distributed publish-subscribe messaging system that maintains feeds of messages in partitioned and replicated topics [77]. All alerts broadcasting modules are producers to Kafka topics, while a consumer was implemented to read the messages from the subscribed topics. When the consumer obtains messages, it also sends them to the correlator for correlation processing.

As Figure 12 depicts, the SMS-DT is built upon a microservices architecture with all components talking to each other using application programming interfaces. The data are fed into each domain analyzer from corresponding loggers and sensors where relevant historical and standard datasets might also be utilized. Each monitoring analyzer has multiple submodules and tools that work on analyzing the live streams of input records, as described in detail in the previous sections. Each module has its own interactive user interfaces that allow the security operator to monitor and control domain-specific conditions. Afterward, when an abnormal behavior is detected in any of the monitored domains, the corresponding analyzer generates a relevant alert and sends it to a Kafka topic. Alerts being broadcasted to Kafka are then consumed by the multidomain awareness module that works on correlating the inter-domain alerts and delivers insights via its own interface about detected security activities in the monitored environment. The technical details and inter-linking between all analyzers were comprehensively described in the previous sections. This cyber resilience mechanism represents an intelligent multi-domain correlator that works to correlate different cyber and physical events received from different data sources on the shop floor. The aim is to provide the user the ability to understand that an attack is occurring through the events being received, as it is important that the user understands what is happening to allow the design of future prevention measures.

For the multi-domain correlator, the SMS-DT platform utilizes its own intelligent correlator (IC) whose function is to correlate the data collected from all previously considered domains. The IC utilized in the platform is a shorter version of the Hybrid Intrusion Detection System presented by Dias et al. [78]. The Hybrid IDS is a highly interpretable and explainable rule-based IDS. The authors’ proposal stands out for its ML support on increasing the knowledgebase by generating new rules. This makes the IDS much more robust and resistant over time. However, the main benefit of using this correlator is its ability to calculate new evidence from the consumed data, along with its intrusion detection capability, which was maintained in our platform. The IC is composed of two different components: drools backend (DB) and correlator interface (CI).

The DB is the correlating component that uses the drools engine [79] to produce inferences using the knowledge base and the calculated dynamic evidences. This component allows users to perform Create, Update, Read, and Delete (CRUD) operations on the rules that compose the knowledge base. It is also automated to perform and persist all correlations made in regard of the collected alerts. As mentioned earlier, all message-producing modules are publishers to the Kafka topic, while the DB is a consumer of the alerts produced by the different domains. This component was developed in Java and uses the Spring Framework to follow Representational State Transfer (REST) architecture. In terms of the internal process, the correlation process occurs in a configured cadence. Every time the DB reaches an execution point, it starts by consuming all alerts published during that time window. Then, the information is filtered in order to obtain a set of valuable pieces of evidence and compute new ones. These forms of evidence are then sent to the drools inference engine, which takes care of correlating the data, along with the rules registered in the knowledge base. When the inference results in a warning or a danger state, a justification is created. This confers the system a high transparency, interpretability, and explainability. The correlation interface is an interactive web-based tool to control the system and provide an overview about the SMS-DT platform and its components. Using a set of dashboards and visualization techniques, it helps operators manage and obtain details about occurring alerts and their correlations in the entire investigated environment.

## 5. Textile Industrial Use-Case

The end-user of this use-case is a company with solutions for textile labels for branding and promotional effects, and of elastane and rubber-coated yarns for several applications in the textile industry. Present in the market for 50 years, the company is a reference in the textile sector and has a multinational cross-sector client portfolio. In a continuous search for improvement, our end-user aims to adopt technological and innovative solutions that can produce a reduction in costs without adversely affecting the performance of the production lines and associated machines.

As in most textile companies, the main assets of our end-user are specialized human resources, Jacquard and Rapier Looms, and cutting machines. The manufacturing resources of the Jacquard and Rapier processes are sensitized, which allows, for example, the calculation of energy consumption. Roughly speaking, the shop floor has five main sections:The weaving section, where the raw product is prepared to feed the Rapier looms.Two different sections with looms, one with Rapier looms and the other with Jacquard looms.A cutting division where the pattern pieces woven on the looms are cut.A final stage where a sequence of different operations is performed to provide the pieces their final commercial appearance.

All these sections have a terminal with a system that is used by the operators to register the production orders. These terminals are also used in the Jacquard and Rapier looms sections to download to a USB flash drive some production order information that needs to be inserted into the looms system. In the two sections with looms, the data of some sensors are collected and sent to the ERP system. In the cutting division, some more recent machines have a Windows production terminal that allows for direct interaction with the local ERP. As can be inferred from the previous description, all the processes in the textile factory have a strong manual component: it is the operators who register in the ERP what is done in the machines; they are also responsible for inserting in the machine the information needed to execute the order, and Jacquard and Rapier looms have connection ports (e.g., USB and serial) that can be used to interact with them. This carries a great risk of incidents because the operator can make (accidentally or not) some mistakes.

This use-case is well suited for our purpose as it is a paradigmatic example of small and medium companies, with both advanced and legacy equipment, and with a long history and simultaneously evolving to the Industry 4.0 model, combining automation and manual work. The FoF needs to have secure-hardened and reliable solutions and it is crucial to empowering these solutions, providing increased efficiency and value to the FoF. Artificial intelligence services bring light to this aim, providing intelligent mechanisms to extract knowledge from the available data.

Thus, in this use case, three different Jacquard looms were completely monitored. Environmental sensors, such as temperature and humidity, which may influence the quality of the final product, were utilized. Induction sensors to monitor loom parameters, number of meters produced, power analyzers, and flowmeters were also used to collect data on the shop floor. The collected information was stored in a data lake that simplified the access to data and the detection of abnormal patterns, resulting in deriving insights about existing and future operations in close-to-real-time scenarios. Furthermore, a set of disruptive intelligent services based on innovative data lake analysis and learning approaches fostered a sustainable, secure, distributed, and efficient set of data knowledge services. Human behavior was also modelled and simulated to investigate stress, fatigue, and lack of attention.

Therefore, using this use-case, we demonstrated the main features that can be delivered to our end-user, which include a set of different data analytics techniques for features extraction, cross-checking manufacturing, human behavior modelling, and context variables. Such techniques support production optimization, increased efficiency, and predictive actions (such as predictive manufacturing). The deliverables also combine data analytics for anomaly detection, as well as advanced decision support system using the data provided by the sensors to ensure a continuous improvement and optimization. Finally, we also delivered machine learning and knowledge-as-a-service models that generate insights and recommendations and guide decision making among the manufacturing system as a whole.

### 5.1. Misuse-Case Scenarios

A misuse-case describes the steps and scenarios that an actor performs in order to accomplish a malicious act against a system or business process. To demonstrate the efficiency and utility of the previously described tools in the detection and remediation of these malicious actions, in the next sections, we describe two misuse-case scenarios on a high level. We decided to have two different types of actors: an operator who makes a mistake that causes a problem in the production line (unintentional attack) and a malicious person who intentionally causes a problem to the company (intentional attack). It is worth noting that we implemented all previously mentioned components and integrated them with the textile shop floor systems. Then, we used them to simulate the scenarios of the following misuse cases in data captured from the real industrial environment.

#### 5.1.1. Misuse-Case A: Unintentional Attack Action

The final product can be affected by an accidental error caused by an operator. The error can be caused by using the wrong material. It can also happen due to an error on the loom that the operator cannot avoid due to their lack of attention, fatigue, or even inexperience. Therefore, in this storyline, the misbehavior of the production process is caused by an underperforming employee that, due to his tiredness level, misbehaves with the work that should be done. The product is then wrongly produced, as detected in the quality assessment stage of the production, forcing the restart of the production line. This storyline is represented in Figure 13.

To prevent such unintentional scenarios, the SMS-DT platform plays a role in detecting abnormal actions via the intervention of three components: the human and energy analyzers and the intelligent correlator.

The human behavior analyzer can detect the tiredness of the operator and raise an alert about the operator’s state. A nearby camera that is responsible for capturing the worker’s behavior is streaming real-time footage. The camera application receives the video stream and sends it to the emotion detector. The emotion detector module classifies the worker’s status on the basis of the streamed pictures and generates an alert due to a reported negative emotion. The alert is reported to the human behavior interface and sent with all relevant information to the corresponding topic in the Kafka data lake. As a preventive measure, the platform might ask the worker to conduct a fatigue test before continuing the work. In terms of energy monitoring, the energy consumption analyzer maintains a regularly updated model for the energy consumption for each machine on the shop floor. It also captures the real-time consumption through the installed loggers. The mistakes that the fatigued or underperforming worker has made lead to an uncommon behavior of some machines or to have some of them completely stopped. During the frequent scanning process, the energy forecasting tool compares the current consumption with the predicted values on the basis of the consumption profile for each machine. Thus, it detects the deviations in the real-time energy values and generates an alert (see Figure 11). Then, it reports the alert to both its interface and the corresponding topic in the Kafka data lake. Afterwards, having the previous two alerts produced to Kafka topics along with other alerts, the drools backend engine in the intelligent correlator consumes all alerts and starts to correlate them. For example, on the basis of the rules stored in its knowledge base, it infers a danger status due to having the previous two alerts triggered at the same period. Accordingly, the engine generates an alert and reports it in the multi-domain awareness dashboard so that it instantly reaches to the security operator who receives it and handles it accordingly. The detailed processes and internal techniques of each component are described in detail in Section 4. Figure 14 shows a generated rule responsible for detecting such a dangerous situation due to abnormal behaviors in both human monitoring and energy domains.

#### 5.1.2. Misuse-Case B: Intentional Attack Scenario

A malicious person who may have access to the looms can intentionally damage the loom, for example, by causing technical problems; or, they can produce more pieces than the ones in the service order, causing economic damages to the company or selling counterfeit products. In this story line (Figure 15), an attacker aims to produce pieces of a renowned end-user’s customer to sell them in the black market. To achieve their goal, the attacker needs the product designs, usually stored in the end-user system, as well as access to the looms to then produce the parts according to the customer’s design. Thus, the attacker starts the attack with two different attacks: one to an end-users’ administrator to obtain privileged access to the administrator computer and find the customer’s designs; the other to an end-user employee to coerce the employee and have access to the looms. After having access to the customer’s designs and coercing the employee, the attacker and the employee access end-user’s facilities to use the looms at unusual hours and produce the pieces to sell them in the black market. Figure 15 represents the different steps of this storyline.

Using the tools described in the previous sections, we can alert about these malicious actions. Figure 16 represents the intervention of our systems into multiple steps of this misuse case, as described in the next paragraph.

All components will orchestrate to detect such a type of attack and alert the security personnel. After the phishing email to an end-user’s administrator, the attacker executes a network scan to find the customer’s designs. The machine-learning-based IDS of the network analyzer monitors the network traffic in real time. Accordingly, it will be able to detect the abnormal access of the company resources, generating an alert and reporting it to the Kafka data lake. On the other hand, the camera application of the human behavior analyzer captures footage of the employee and sends it to the emotion detector, which in turn identifies the employee’s negative emotions resulting from the attacker’s coercion and reports corresponding alerts to the Kafka topic. Additionally, using the looms at unusual hours will be detected by the forecasting tool, which monitors the real-time consumption and compares it to the loom’s regular profile. As a result, an energy consumption alert is also created and sent to the data lake. Figure 17 shows the three mentioned alerts generated by SMS-DT systems as inspected in the intelligent correlation interface. With all these alerts being received by the intelligent correlator at the same time, they will be intelligently correlated in a similar manner to the previous misuse case. The resulting correlation alert with all the details can be instantly seen in the multi-domain awareness dashboard. The detailed mechanism and models of how each component works is comprehensively described in Section 4. Finally, Figure 18 depicts some danger and warning alert correlations as inspected in the intelligent correlation interface.

## 6. Conclusions

The inevitable transition from traditional industrial environments into modern factories has been accompanied by increasing connectivity between cyber and physical components. Realizing smart factories requires advanced models to simulate and control the combination of various complex systems that were not originally designed to operate in a unified architecture. Despite this combination resulting in optimized production processes, the introduction of newly developed techniques compromises the overall system to a wide range of security and safety concerns. To this end, having a centralized mechanism to monitor all industrial modules is very necessary, not only to have an overall overview of the investigated environment, but also to correlate events occurring simultaneously in different parts of the system.

Our SMS-DT platform was developed to capture attacks, vulnerabilities, and safety issues on the basis of intelligent correlation of messages from multiple domains in an industrial cyber-physical environment. Relying on the digital twin approach and intelligent services, the SMS-DT utilizes environmental sensors, network monitoring, and workers behavior analysis models to deliver real-time multi-domain situational awareness in smart factories. Nevertheless, regardless of their operational complexity, most modern factories utilize heterogeneous and highly integrated systems, most of which deals with new technologies that might be not yet stable to run faultlessly. This makes it difficult even for small- and medium-sized factories to secure proper continuous productions. For this reason, our SMS-DT platform has been developed to fulfill this requirement. Nonetheless, there are still many opportunities for future approaches to build upon our platform to investigate this gap and bridge it accordingly. Future activities might focus on discovering further domains other than the three ones covered in this study. For example, researchers can integrate IoT sensors and other embedded systems to monitor environmental conditions on the shop floor, such as the internal temperature, workers/machines movement, functional alarms produced by machines, and access control to restricted industry departments. Additionally, with the SMS-DT platform overloaded with numerous AI-based systems, there is a need to assess the combined computational cost and proper techniques for all included machine learning models while working together. The role of cloud-based architectures to handle this load can also be investigated. This assessment does not include only the performance of prediction models, but also the robustness and explainability of such models to correlate alerts and generate new ones. Moreover, considering the continuous evolution of machine learning models, the investigation of better deep learning and neural network models for both alerts detection and correlation is also an open area for research, taking into account the most efficient techniques to generate insights in real-time or near real-time conditions.

Finally, we strongly believe that the detection of abnormal or dangerous situations based on alert patterns from multiple domains can generate sharable insights, as the same pattern that is discovered in specific industrial situations might apply to multiple other industries. This implies the need for security specialists and other stakeholders and policymakers to define frameworks and standards to share such re-usable insights in order to protect the overall industrial community while keeping the sensitive details for each use case securely maintained.

## Figures and Tables

**Figure 1 sensors-22-09915-f001:**
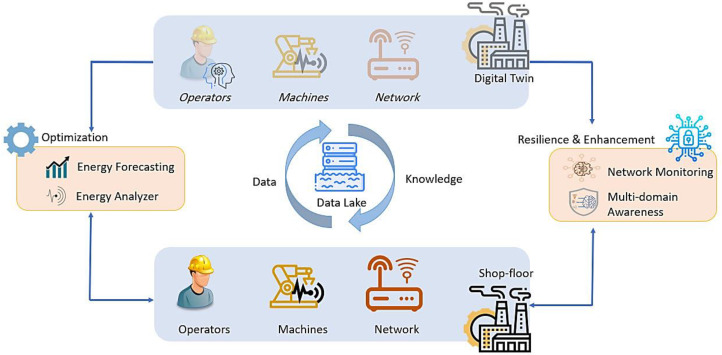
Key capabilities of the holistic security and safety architecture.

**Figure 2 sensors-22-09915-f002:**
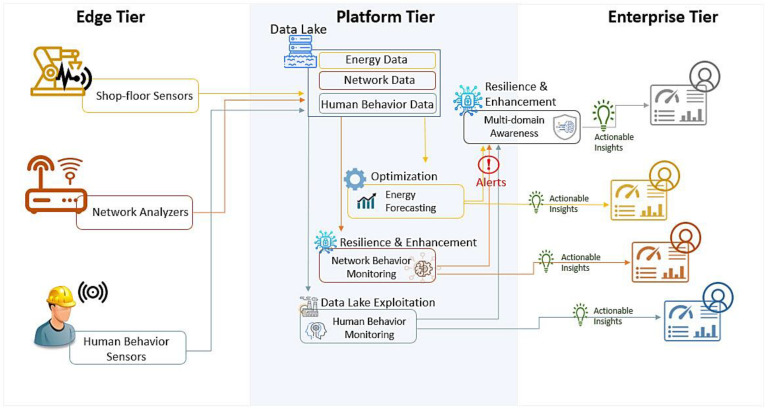
SMS-DT platform architecture.

**Figure 3 sensors-22-09915-f003:**
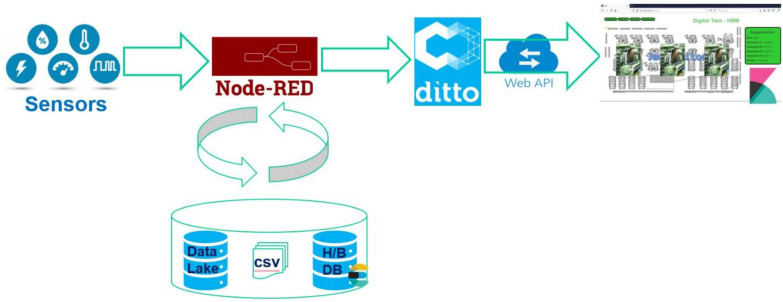
SMS-DT digital twin overview.

**Figure 4 sensors-22-09915-f004:**
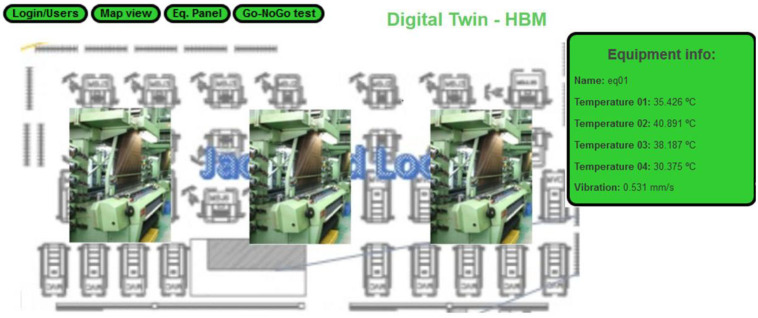
SMS-DT map view.

**Figure 5 sensors-22-09915-f005:**
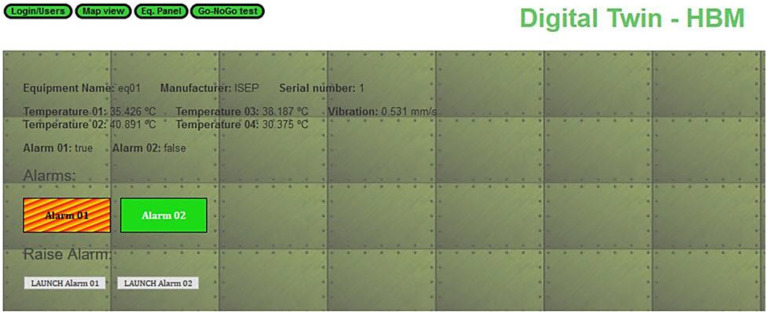
SMS-DT equipment panel info.

**Figure 6 sensors-22-09915-f006:**
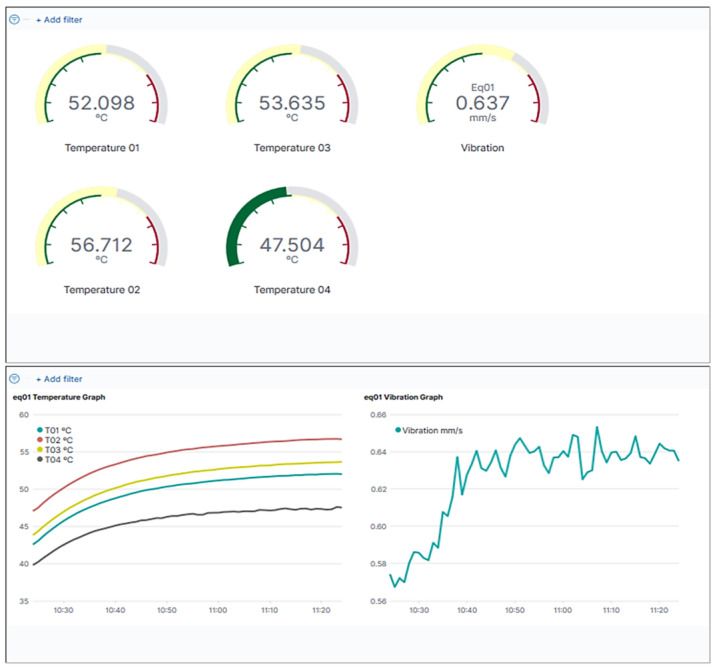
SMS-DT equipment panel gauges.

**Figure 7 sensors-22-09915-f007:**
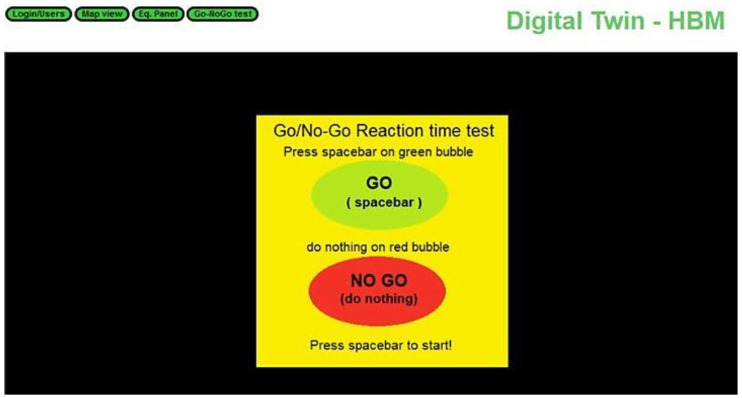
SMS-DT Go/NoGo interface.

**Figure 8 sensors-22-09915-f008:**
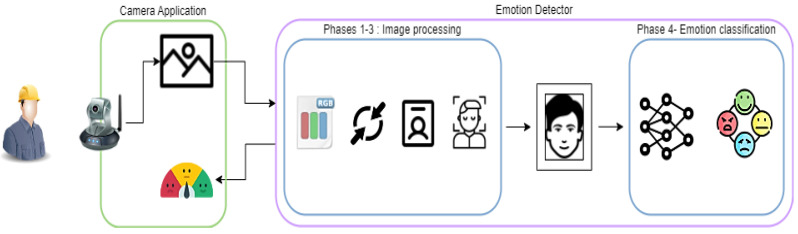
Human Behavior Analyzer workflow.

**Figure 9 sensors-22-09915-f009:**
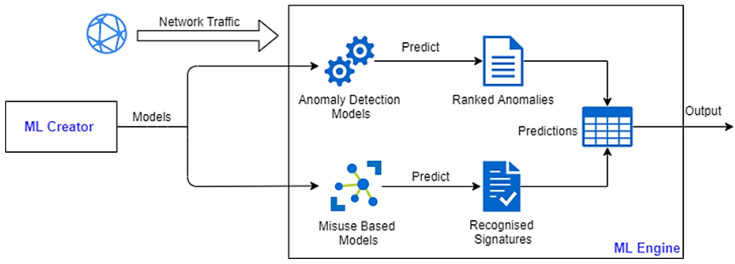
SMS-DT machine learning engine.

**Figure 10 sensors-22-09915-f010:**
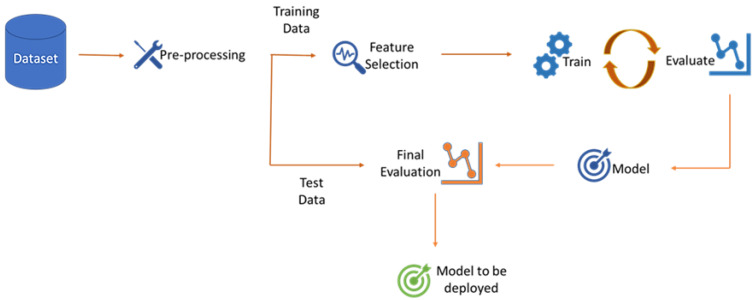
SMS-DT ML creator.

**Figure 11 sensors-22-09915-f011:**
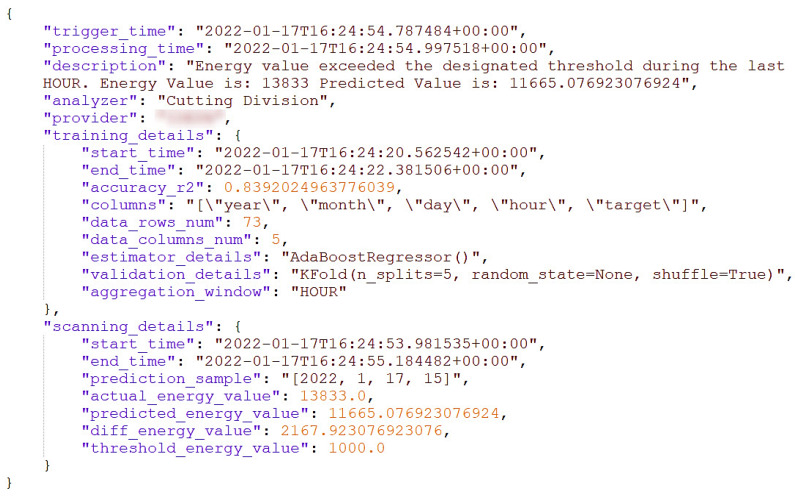
Alert triggered due to abnormal energy consumption.

**Figure 12 sensors-22-09915-f012:**
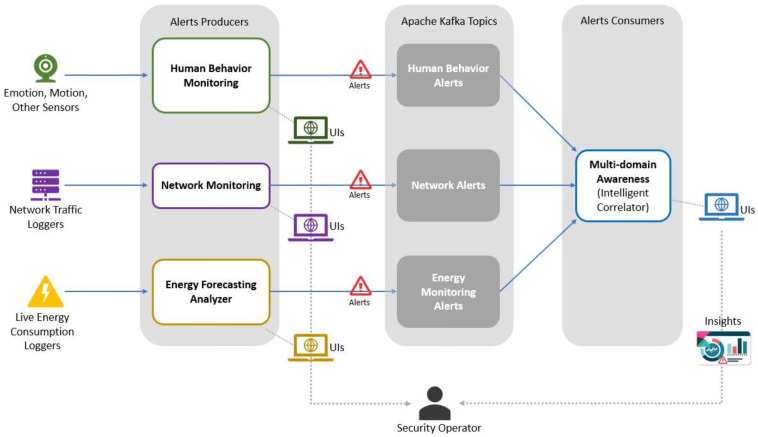
Correlation of alerts from multiple domains.

**Figure 13 sensors-22-09915-f013:**
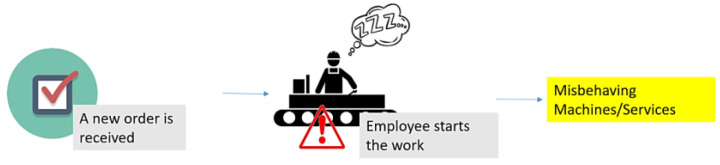
Misuse-case A: unintentional attack action.

**Figure 14 sensors-22-09915-f014:**
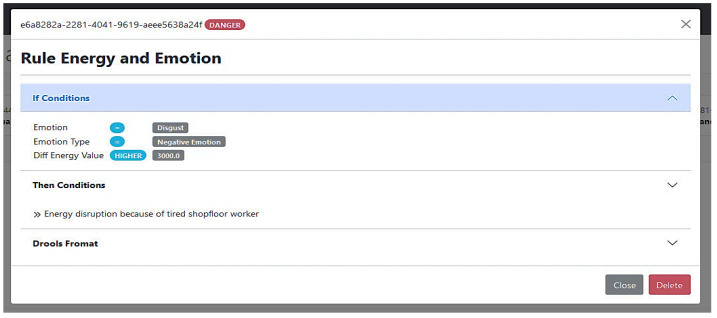
Negative emotions and energy deviations rule generated by SMS-DT systems.

**Figure 15 sensors-22-09915-f015:**
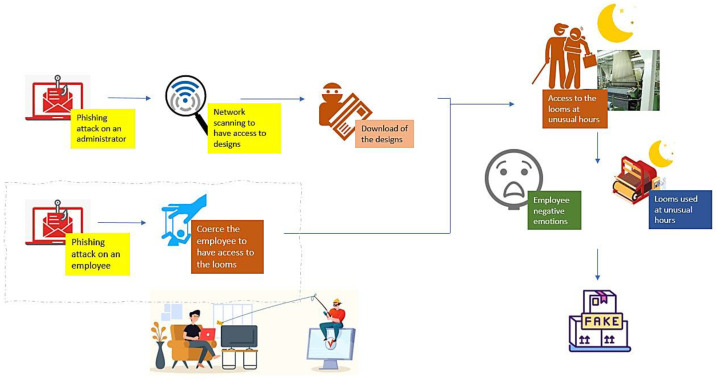
Misuse-case B: intentional attack scenario.

**Figure 16 sensors-22-09915-f016:**
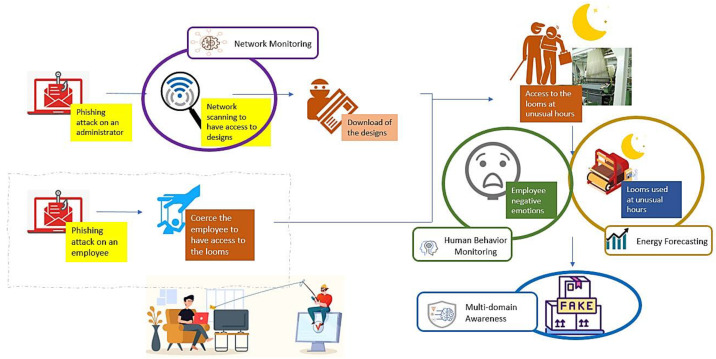
Misuse-case B: intentional attack scenario with SMS-DT detection solutions.

**Figure 17 sensors-22-09915-f017:**
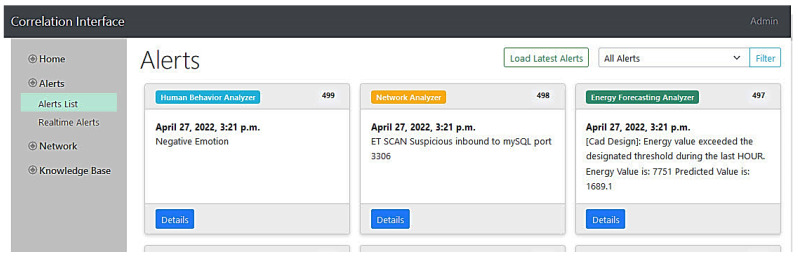
Alerts inspected in the correlation interface.

**Figure 18 sensors-22-09915-f018:**
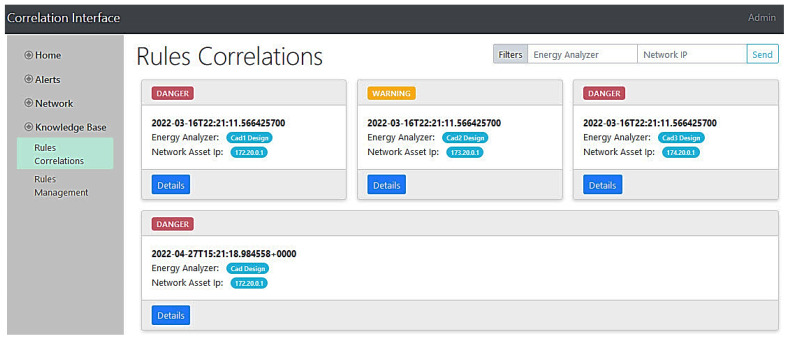
Alert correlations in the correlation interface.

## Data Availability

Not applicable.

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
