# Peer review of "Holistic Security and Safety for Factories of the Future"

_sensors, 2022, doi:10.3390/s22249915_

Round 1
Reviewer 1 Report
This manuscript focuses on improving the safety operation of factories, via the combination of sensors and artificial intelligence. It looks more like a review rather than a research paper. The authors described a very comprehensive approach but did not present any results to demonstrate the approach can work. All figures are schematics and some of them are blurred. I suggest reconsidering a new version of this paper, provided that the authors can present results to show the performance of their approach and compare it with others in literature.
Reviewer 2 Report
The manuscript proposes this SMS-DT platform aimed to simulate and monitor industrial conditions in a digital twin-based architecture.
The study performed by the authors is relevant, interesting and, in general, the effort made is really appreciated.
Moreover, the study can also be aplied to oteher tematics and use cases.
The paper il well articulated and the concepts expressed are clear and well proposed.
Most of the content of the paper seems presented in a descriptive fashion, without going too much into the technical details.
The authors shall to improve this part by better definig the adopted machine learning techniques and models applied to the use case.
Also the description of the platform need to be improved. In particular, as it is clear the platform architecture model, it is not clear how those components intercat among them, neither the infrastructure used to deploy such components.
Another important point that shall be improved is the investigation of the outcomes of running and past projects, focused both in the study of DT and in the definition of related platforms as a support of the DT models for industry-related appications.
During the past decade, in fact, an important work has been done to define and made avialable cloud-based tools and services for (but not limeted to) the hybrid DT.
As an example, the activity carried out by the IOTWINS project (https://cordis.europa.eu/project/id/857191/reporting) should be taken into consideration and properly cited within the manuscript (Costantini, A.; Di Modica, G.; Ahouangonou, J.C.; Duma, D.C.; Martelli, B.; Galletti, M.; Antonacci, M.; Nehls, D.; Bellavista, P.; Delamarre, C.; Cesini, D. IoTwins: Toward Implementation of Distributed Digital Twins in Industry 4.0 Settings. Computers 2022, 11, 67. https://doi.org/10.3390/computers11050067)
Moreover, the conclusion describes what has been done without presenting future activities nor improvements.
In such respect, authors shall take into account the evolution of the AI and machine learning techniques and models, including the increasing adoption of specialized hardware (e.g. GPUs) that, for neural network and deep learning computing models in particular, can really speed-up the computation and introduce an innovative approach to the study of DT models.
Reviewer 3 Report
The manuscript is well developed and well in the journal scope can be considered as the area of interest for readers. However few minor changes are required in manuscript before publication:
1. Abstract needs revision. Few major novel results can be added in the abstract in a crisp manner to improve the quality of manuscript and readability.
2. Keywords are not sufficient. Some other keywords related to study such as smart factory can be added.
3. Introduction is well written but no research questions or objectives have been defined which needs to be added in the Introduction part.
4. The study is not supported by latest literature published in this area. Few more latest studies from leading journals can be added in the literature part to support the literature such as:
a. The future of factories: different trends
b. Human–robot collaboration trends and safety aspects: A systematic review
c. Smart industrial robot control trends, challenges and opportunities within manufacturing
d. Industry 4.0 technologies for manufacturing sustainability: a systematic review and future research directions
e. Future industrial networks in process automation: Goals, challenges, and future directions
5. Contribution of authors is well presented however figure 11 is not clear it is better to provide and present high quality image for Figure 11.
6. Implications related to study are missing in the work. It would be better to include implications for practitioners and policymakers.
7. Conclusion part can be revised by adding few more future scopes related to study in the sustainability context.
The manuscript can be accepted after minor changes for publication in journal.
Round 2
Reviewer 1 Report
My comments have been addressed and the manuscript has been seriously improved.